# Toxoplasmosis infection among pregnant women in Africa: A systematic review and meta-analysis

**Tamirat Tesfaye Dasa**[1]*, **Teshome Gensa Geta**[2], **Ayalnesh Zemene Yalew**[3], **Rahel Mezemir Abebe**[3], **Henna Umer Kele**[4]

**1** Hawassa University, College of Medicine and Health Sciences, Hawassa, Ethiopia, **2** Department of Biomedical Science, Wolikite University, College of Health Sciences and Medicine, Wolikite, Ethiopia, **3** Saint Paul's Hospital Millennium Medical College, Addis Ababa, Ethiopia, **4** Malawi University, College of Medicine, Zomba, Malawi

* tamirathenna@gmail.com

## Abstract

The epidemiology of toxoplasmosis in pregnancy is a major issue in public health. Toxoplasmosis is caused by the protozoan parasite. Toxoplasma parasite is at high risk for life-threatening diseases during pregnancy. Congenital toxoplasmosis results from a maternal infection acquired during gestation. Therefore, this systematic review and meta-analysis was aimed to determine the seropositive prevalence of toxoplasmosis infection among pregnant women who attended antenatal care in a health facility in Africa. A systematic review and meta-analysis of published and unpublished studies were included. Databases such as MEDLINE, PubMed, EMBASE, CINAHL, Web of Science, African Journals Online were used with relevant search terms. The quality of the articles was critically evaluated using the tool of the Joanna Briggs Institute. Data were extracted on Microsoft word 2016. Meta-analysis was conducted using STATA 14 software. The heterogeneity and publication bias were assessed using the $I^2$ statistics and Egger's test, respectively. Forest plots were used to present the pooled prevalence and odds ratio with a 95% confidence interval of meta-analysis using the random effect model. In total, 23 studies comprising 7,579 pregnant women across ten countries in Africa were included in this meta-analysis. The overall prevalence of seropositive toxoplasmosis among pregnant women in Africa was 51.01% (95% CI; 37.66, 64.34). The heterogeneity test showed that heterogeneity was high, $I^2$ = 99.6%, P-value < 0.001. The variables responsible for the source of heterogeneity were included from Cameroon, the Democratic Republic of Congo, and Ethiopia. Overall prevalence of toxoplasmosis in Africa showed that more than one-half of pregnant women were infected. The risk of acquiring toxoplasmosis infection during pregnancy is high; hence, preventive measures to avoid exposure of pregnant women to Toxoplasma infection should be strictly applied.

**Data Availability Statement:** All relevant data are within the manuscript and its Supporting Information files.

**Funding:** We have not received any financial support for this work.

**Competing interests:** We have declared that no competing interests exist.

**Abbreviations:** ANC, Antenatal Care; MeSHs, Medical Subject Headings; OR, Odds Ratio; DR, Democratic Republic of Congo; PRISMA, Preferred Reporting Items for Systematic Reviews and Meta-analysis.

## Introduction

Toxoplasmosis is caused by the protozoan parasite, Toxoplasma gondii. Globally, it accounts for over 60% of the populations that have been infected with Toxoplasma [1]. Toxoplasma gondii infects a large proportion of the world's human population. Toxoplasma parasite causes life-threatening diseases like immunological impairments and congenital infection of the foetus. Congenital toxoplasmosis results from a maternal infection acquired during gestation [2]. The prevalence of toxoplasmosis infection in pregnant women of vertical via transmission in Brazil was 68.37%, which was suspected by IgM antibodies detection in the peripheral blood [3]. An effective vaccine for use in humans, while serving to reduce mortality and morbidity associated with infection, would also have economic benefits, as it would reduce the financial burden of lifelong care needed for those with severe chronic diseases [4]. Unborn babies are at high risk of being infected with toxoplasmosis during pregnancy. On average, 4 out of 10 of such infections do pass from pregnant women to their babies [5].

Healthy individuals who become infected with Toxoplasma gondii rarely develop any symptoms because their immune system keeps the parasite from causing illness. About 95% of the population infected with Toxoplasma did not develop any symptoms. If a woman becomes newly infected with Toxoplasma during pregnancy, she can pass the infection to her unborn baby. Toxoplasmosis infection during pregnancy has many adverse effects, such as miscarriage, stillbirth, or damage to the baby's brain and other organs, particularly the eyes. However, most babies born with toxoplasmosis have no obvious damage at birth but develop symptoms, usually eye damage or potential vision loss, during childhood or even adulthood, mental disability, and seizures [5, 6].

The toxoplasmosis in pregnant women was found to be high as a result show through many observational studies in Africa. However, toxoplasmosis is still underestimated as the burden of health and it was also not yet known the overall prevalence in Africa. Awareness about the potentially overwhelming sequelae of toxoplasmosis infection during pregnancy remains low, even in countries with a high burden of toxoplasmosis. Estimation of the prevalence of toxoplasmosis infection in pregnant women at the continent level can help increase its awareness among healthcare policymakers and help develop guidelines to address this serious public health issue, including implementation of prenatal screening and treatment programs for Toxoplasma infection. Therefore, this systematic review and meta-analysis aimed to determine the seropositive prevalence of toxoplasmosis infection among pregnant women who attended antenatal services in Africa.

## Materials and methods

### Study protocol

A systematic review and meta-analysis of published and unpublished studies were conducted to identify the pooled prevalence of toxoplasmosis among pregnant women in Africa. We reported this systematic review and meta-analysis using the Preferred Reporting Items for Systematic Reviews and Meta-Analyses (PRISMA) report 2009 Checklist [7], and that the checklist was strictly followed (**S1 File**).

### Eligibility criteria

We included observational studies (cross-sectional and cohort) that have been conducted in health facilities in different regions of Africa on the prevalence of toxoplasmosis among pregnant women while attending antenatal care units. Also, studies published and accessible until March 30, 2020, and written in the English language were eligible for extract. Articles with

irretrievable full texts (after requesting full texts from the corresponding authors via email and/or Research Gate), records with unrelated outcome measures, and articles with missing or insufficient outcomes were excluded. Reviews, Case-control, commentaries, editorial, case series/reports, and patient stories were also excluded from the systematic review.

## Source of information

We searched databases such as MEDLINE, PubMed, EMBASE, CINAHL, Web of Science, Scopus, African Journals Online (AJOL), Google, Google Scholar, World Cat, Research Gate, and Mednar. Advanced search strategies were applied in major databases to retrieve relevant findings closely related to the prevalence of toxoplasmosis of pregnant women.

## Searching strategies

The search was conducted with the aid of carefully selected keywords and indexing terms. The search strategy *included "Toxoplasma gondii" OR "toxoplasmosis" AND Pregnant Women OR Antenatal care AND Africa,"* and was used distinctly and in combination using the Boolean operators like "OR" or "AND". In the search, the above search terms were joined with the name of all countries included in Africa. The overall search result was compiled using End-Note X8 citation manager software [8] (**S2 File**).

## Study selection process

All searched articles were exported to the EndNote X8 citation manager and duplicated studies were removed. Studies were screened through careful reading of the title and abstract. The two authors (TT and TG) screened and evaluated studies independently. The titles and abstracts of studies that mentioned the outcomes of the interest (Toxoplasmosis/ pregnant women/ Africa) were considered for further evaluation. The full text of the studies was further evaluated based on objectives, methods, participants/population, and key findings. The three authors (AZ, RM, and HU) independently evaluated the quality of the studies against the checklist. The discrepancy for the inclusion of articles was resolved through discussion, and by consulting an expert. The overall study selection process is presented using the PRISMA statement flow diagram [9] (**Fig 1**).

## Data extraction and quality assessment

After selecting the appropriate articles, data were extracted by two investigators independently (TG and RM) using a data extraction template and presented using Microsoft word 2016 (containing author & year, setting, study design, sample size, study subject, data collection methods, the primary outcome of interest (**Table 1**). The accuracy of the data extraction was verified by comparing the results with the data extraction by the second group of investigators (TT, AZ, and HU), who independently extracted the data in a randomly selected subset of papers (30% of the total). The quantitative data were extracted from the included articles and summarised in a Microsoft Excel sheet. For the prevalence of toxoplasmosis infection, we used unconverted proportional data to calculate the proportion/prevalence of toxoplasmosis infection in percentage using Stata version 14. The quality of the included articles was critically evaluated using the quality assessment tool for observational studies (cross-sectional and cohort studies) developed by the Joanna Briggs Institute (JBI) [10]. The two groups of authors (TT and TG) and (AA, RH, and HU) independently evaluated the quality of the studies. The mean score of the two groups of authors was taken for a final decision. The differences in the inclusion of the studies were resolved by consensus. The cross-sectional studies checklist was graded out of 8 points, and the cohort studies checklist was graded out of 11 points. The

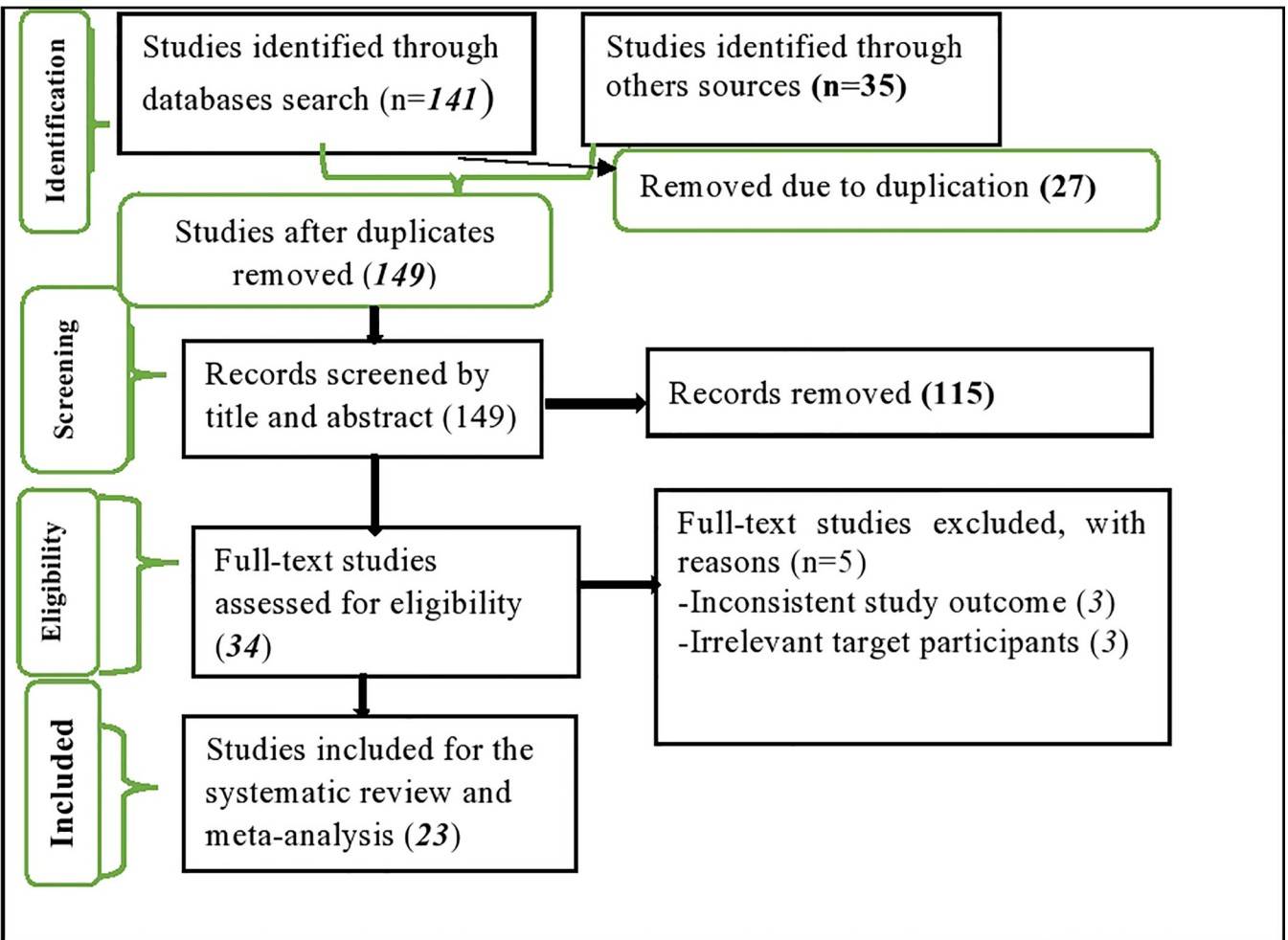

**Fig 1. PRISMA flow diagram of the studies included in the meta-analysis.**

included studies were evaluated against each indicator of the tool and categorised as high-, moderate-, and low quality. Studies with a score greater than or equal to 60% were included. This critical appraisal was conducted to assess the internal validity (systematic error) and external validity (generalisability) of studies and to reduce the risk of biases.

## Heterogeneity and publication bias

The heterogeneity test of included studies was assessed using the $I^2$ statistics. The p-value for $I^2$ statistics less than 0.05 was used to determine the presence of heterogeneity. Low, moderate, and high heterogeneity was assigned to $I^2$ test statistics results of 25, 50, and 75%, respectively [11]. The publication bias was assessed using the Egger regression asymmetry test [12, 13] for meta-analysis results which showed the presence of publication bias (Egger test = p < 0.05).

## Strategy for data synthesis

The raw numerical data and total sample size from each study were extracted and recorded on Microsoft word and then exported to an Excel Spreadsheet. Meta-analysis was conducted using STATA 14 software to compute the pooled prevalence of toxoplasmosis among pregnant

**Table 1. Description of study participants and characteristics of studies included in the systematic review and meta-analysis.**

| No. | Authors & years | Countries | Design of the study | Serological analysis test done | sample size | Prevalence (%) | The primary outcome |
|---|---|---|---|---|---|---|---|
| 1. | (Abamecha and Awel 2016) [15] | Ethiopia | cross-sectional | Enzyme-linked immunosorbent assay. | 232 | 85.30 | Pregnant women following antenatal care |
| 2. | (Agmas, Tesfaye, et al. 2015) [16] | Ethiopia | cross-sectional | Latex agglutination test | 263 | 68.4 | Seroprevalence of Toxoplasma gondii infection among pregnant women |
| 3. | (Awoke, Nibret, et al. 2015) [17] | Ethiopia | cross-sectional | Latex agglutination test | 384 | 18.5 | Seroprevalence Toxoplasma gondii infection in pregnant women |
| 4. | (Bamba, Cissé, et al. 2017) [18] | Burkina Faso | cross-sectional | Enzyme-linked immunosorbent assay. | 316 | 31.1 | Seroprevalence toxoplasma gondii infection in pregnant women |
| 5. | (De Paschale, Ceriani et al. 2014) [19] | Benin | cross-sectional | Enzyme-linked fluorescent assay. | 283 | 30.0 | Antenatal screening for Toxoplasma gondii |
| 6. | (Doudou Yobi, Renaud Piarroux, et al. 2014) [20] | DR. Congo | cross-sectional | Enzyme-linked fluorescent assay. | 781 | 80.3 | Toxoplasmosis among pregnant women |
| 7. | (Endris, Belyhun, et al. 2014) [21] | Ethiopia | cross-sectional | Latex agglutination test | 385 | 88.6 | Seroprevalence of Toxoplasma gondii in Pregnant Women |
| 8. | (Fenta 2019) [22] | Ethiopia | cross-sectional | Enzyme-linked immunosorbent assay. | 494 | 81.8 | Seroprevalence of Toxoplasma gondii among pregnant women |
| 9. | (Frimpong, Makasa, et al. 2017) [23] | Zambia | cross-sectional | Rapid test cassettes by CTK Biotech | 411 | 5.87 | Seroprevalence of toxoplasmosis in pregnant women |
| 10. | (Gelaye, Kebede, et al. 2015) [24] | Ethiopia | cross-sectional | Latex agglutination test | 288 | 85.4 | Prevalence of Toxoplasma gondii in pregnant women |
| 11. | (Koffi, Konaté, et al. 2015) [36] | Ivory Coast | Prospective | Enzyme-linked fluorescent assay. | 385 | 58.70 | Seroepidemiology of Toxoplasmosis in Pregnant Women |
| 12. | (Linguissi, Nagalo, et al. 2012) [37] | Burkina Faso | Retrospective | Enzyme-linked immunosorbent assay. | 182 | 20.3 | Seroprevalence of toxoplasmosis in pregnant women |
| 13. | (Murebwayire, Njanaake, et al. 2017) [25] | Rwanda | cross-sectional | Enzyme-linked immunosorbent assay. | 384 | 9.6 | Seroprevalence of Toxoplasma gondii infection among pregnant women |
| 14. | (Mwambe, Mshana, et al. 2013) [26] | Tanzania | cross-sectional | Enzyme-linked immunosorbent assay. | 350 | 30.9 | Sero-prevalence Toxoplasma gondii infection among pregnant women |
| 15. | (Nasir, Aderinsayo, et al. 2015) [27] | Nigeria | cross-sectional | Enzyme-linked immunosorbent assay. | 360 | 48.9 | Prevalence of Toxoplasma gondii Antibodies among Pregnant Women |
| 16. | (Negero, Yohannes, et al. 2017) [28] | Ethiopia | cross-sectional | Latex agglutination test | 210 | 75.7 | Seroprevalence of T. gondii infection in pregnant women |
| 17. | (Njunda, Assob, et al. 2011) [29] | Cameroon | | Enzyme-linked immunosorbent assay. | 110 | 72.73 | Seroprevalence of Toxoplasma gondii infection among pregnant women |
| 18. | (Olusi, Grob, et al. 1996) [30] | Nigeria | cross-sectional | Enzyme-linked immunosorbent assay. | 606 | 43.7 | High Incidence of Toxoplasmosis During Pregnancy |
| 19. | (Paul, Kiwelu, et al. 2018) [31] | Tanzania | cross-sectional | Enzyme-linked immunosorbent assay. | 254 | 44.5 | Toxoplasma gondii seroprevalence among pregnant women |
| 20. | (Rodier, Berthonneau, et al. 1995) [32] | Benin | cross-sectional | Enzyme-linked immunosorbent assay. | 211 | 53.6 | Seroprevalences of toxoplasma among pregnant women |
| 21. | (Simpore, Savadogo, et al. 2006) [33] | Burkina Faso | cross-sectional | Enzyme-linked immunosorbent assay. | 129 | 20.2 | Toxoplasma gondii among HIV-Negative Pregnant Women |
| 22. | (Teweldemedhin, Gebremichael, et al. 2019) [34] | Ethiopia | cross-sectional | Enzyme-linked immunosorbent assay. | 360 | 35.6 | Seroprevalence of Toxoplasma gondii among pregnant women |
| 23. | (Zemene, Yewhalaw, et al. 2012) [35] | Ethiopia | cross-sectional | Enzyme-linked immunosorbent assay. | 201 | 83.6 | Seroprevalence of Toxoplasma gondii among pregnant women |

women. Forest plots were used to show the magnitude of toxoplasmosis among pregnant women in Africa. A meta-analysis of observational studies was conducted, based on recommendations made by Higgins et al. (An $I^2$ of 75/100%, suggesting considerable heterogeneity). In the meantime, heterogeneity between the included studies was examined using the $I^2$ statistic [14]. Therefore, the presence of heterogeneity between studies was assumed if the $I^2$ statistic

is greater than 75% and the p-value less than 0.05. A random-effects model [11] was used to determine the pooled prevalence of toxoplasmosis among pregnant women for antenatal care. Subgroup analyses and meta-regressions were also performed to verify the source of heterogeneity in the studies used in the systematic review.

# Results

## Study selection

This systematic review and meta-analysis included published and unpublished studies conducted on toxoplasmosis infection among pregnant women in Africa. A total of 176 articles were identified through the major electronic databases and other relevant sources. From all identified studies, 27 articles were removed because of duplication, while 149 studies were reserved for further screening. Of these, 115 were excluded after being screened according to titles and abstracts. Of the 34 remaining articles, 11 studies were excluded because of inconsistency with the inclusion criteria set for the review. Finally, 23 studies that fulfilled the eligibility criteria were included for the systematic review and meta-analysis. The general characteristics and descriptions of the studies selected for the meta-analysis are outlined in (**Fig 1**).

## Characteristics of included studies

In this review, 10 studies from African countries were included. All studies included in the final analysis were conducted in a health facility setting. In brief, out of the 23 studies as shown in Table 1; 21 studies were cross-sectional studies [15–35] while 2 studies followed a prospective cohort study design [36, 37]. The serological analysis was done to check the presence of anti-Toxoplasmosis gondii antibodies. While 15 studies used enzyme-linked immunosorbent assay [15, 18, 22, 25–35], 1 study used the Combo Rapid test cassettes by CTK Biotech [23]. Also, 5 studies used latex agglutination test [16, 17, 21, 24, 28], and 2 studies used enzyme-linked fluorescent assay [20, 36]. Almost all included studies were conducted in three African regions: 2 studies from Central Africa [20, 29], 8 studies from West Africa [18, 19, 27, 30, 32, 33, 36, 37] and 13 studies from East Africa [15–17, 21–26, 28, 31, 34, 35]. The total sample size of the included studies ranged from a minimum of 110 in a study conducted in Cameroon [29] to a maximum of 781 in a study conducted in the Democratic Republic of Congo [20]. Overall, this review included 7, 579 study participants, and out of these, 3901 (51.14%) pregnant women were infected by toxoplasmosis gondii (**Table 1**).

## Prevalence of toxoplasmosis of pregnant women in Africa

A total of 23 studies that reported the prevalence of toxoplasmosis among pregnant women in Africa were included in this meta-analysis [9–31]. The studies included in a meta-analysis from different African countries. The prevalence of toxoplasmosis reported among pregnant women ranged from 5.87% in Zambia [17] to 88.60% in Ethiopia [15]. Moreover, the overall pooled prevalence of toxoplasmosis among pregnant women in Africa was 51.01% (95% CI; 37.66, 64.34). The heterogeneity test showed the presence of heterogeneity, $I^2$ = 99.6%, P-value < 0.001 (**Fig 2**). However, Egger's regression asymmetry test also detected a significant publication bias. Moreover, the funnel plot (**Fig 3**) and the Egger regression test result showed that the intercept ($B_0$) was 0.21 (95% CI, 0.18 to 0.24), with the 2-tailed P-value < 0.0001. After adjustment, the final pooled prevalence of toxoplasmosis among pregnant women in Africa after the trim and fill analysis was 51.01% (95% CI; 37.682, 64.338) (**Fig 2**).

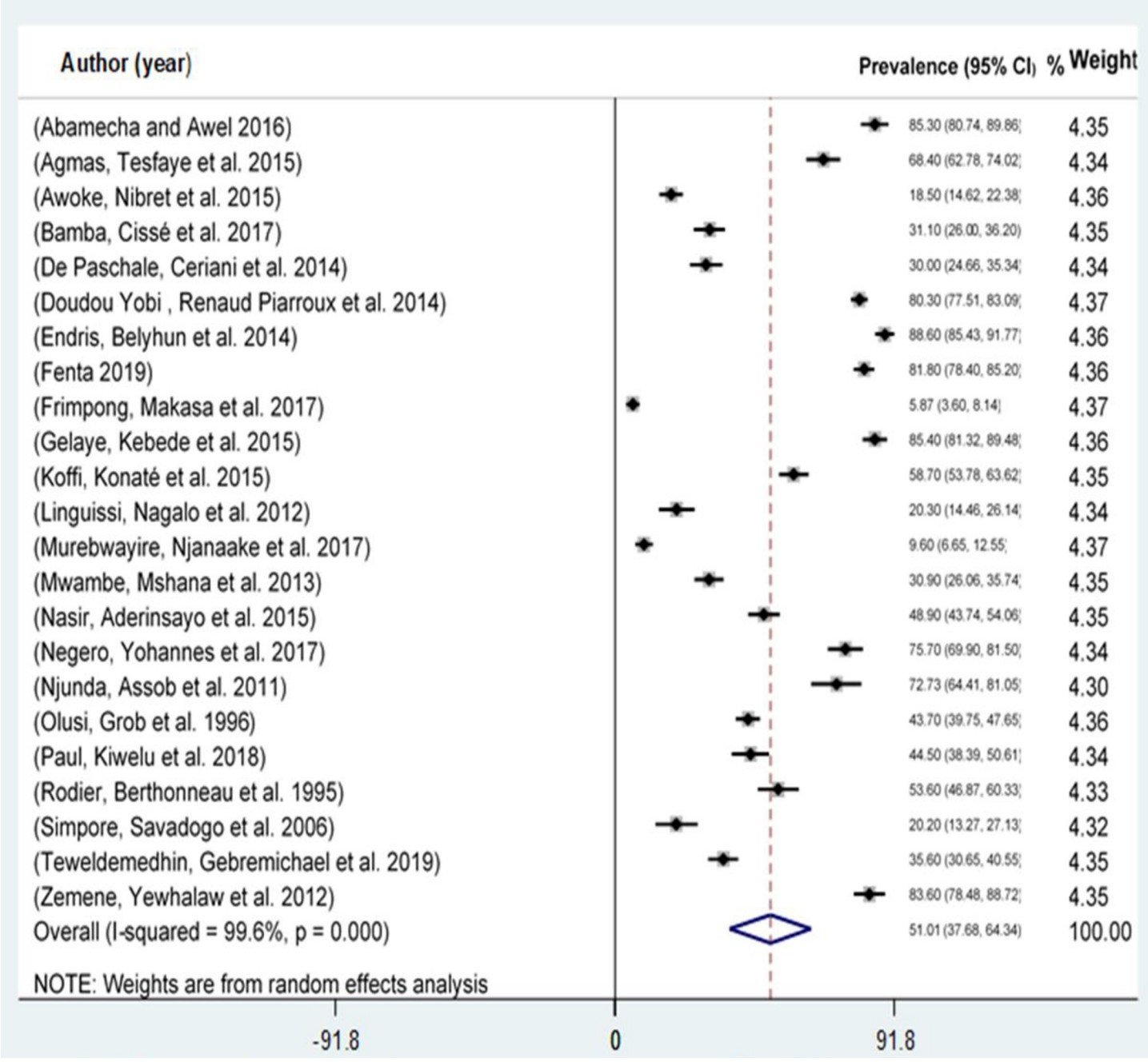

**Fig 2. Pooled prevalence of toxoplasmosis among pregnant women in Africa.**

## Risk of bias across studies

**Subgroup analysis.** Subgroup analyses were done by parameters such as countries, sub-African regions, serological analysis tests, and study design. The analysis done using country showed that the highest prevalence of toxoplasmosis among pregnant women was 80.30% (95% CI: 77.51, 83.09) in the Democratic Republic of Congo, followed by 69.21% (95% CI: 51.94, 86.49) in Ethiopia and the lowest in Zambia, 5.87% (95% CI:3.60, 8.14). However, only a

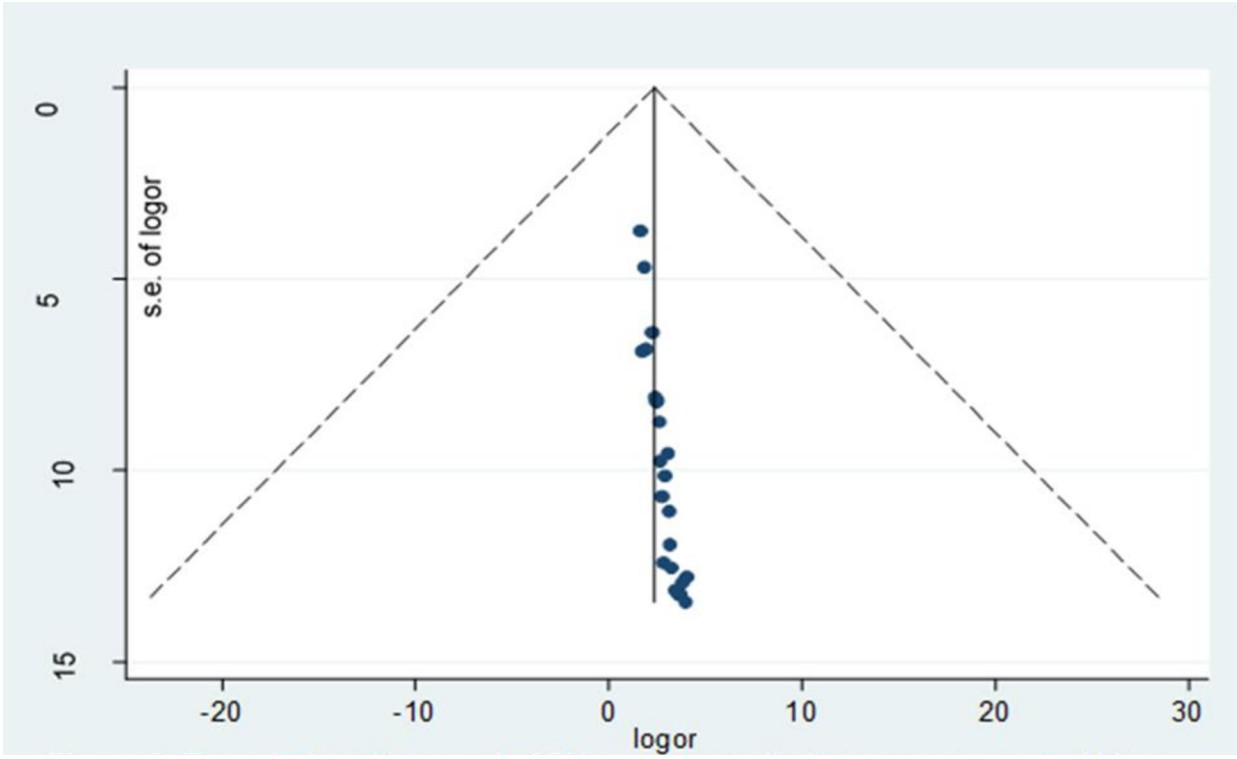

**Fig 3. Funnel plot with pseudo 95% confidence limits for bias assessment.**

single study was considered from DR. Congo and Zambia. On other hand, subgroup analysis done using the sub-African regions showed the highest prevalence in Central Africa, 77.57% (95% CI: 70.45, 84.70), followed by East Africa 54.89% (95% CI: 34.64, 75.14). Subgroup analysis did not show any significant differences between the respective groups, as indicated by overlapping 95% CIs, except for subgroups based in Central Africa and Nigeria (**Table 2**).

## Meta-regression analysis for sources of heterogeneity

A meta-regression analysis was conducted to detect the possible source of heterogeneity found with a high P-value of less than 0.05. This subgroup meta-regression analysis was necessary since heterogeneity of pooled prevalence is high enough to affect the interpretation of findings. The analysis was aimed to identify the source of heterogeneity ensure correct interpretation of the findings. Again, this analysis has shown significant heterogeneity among countries and serological tests, but none among the sub-African region and study design. The possible source for heterogeneity was included from Cameroon, DR. Congo, Ethiopia, and serological analysis done using enzyme-linked immunosorbent assay; hence, these four variables were taken to be responsible for the source of the heterogeneity. There was no statistical significance in subgroup levels of the sub-African region and study design (**Table 3**).

## Discussion

This comprehensive systematic review and meta-analysis were conducted to provide reliable information on toxoplasmosis infection among pregnant women. The authors used extensive and comprehensive search strategies from several databases and included published and unpublished studies and gray literature. Studies were appraised for quality methodology using

**Table 2. Subgroup analyses for the prevalence of toxoplasmosis pregnant women in Africa.**

| Subgroup | No_ of included studies | Total sample size | Prevalence OR (95% CI) | Heterogeneity statistics | | |
|---|---|---|---|---|---|---|
| | | | | Tau-Squared | $I^2$ | p-value |
| **By countries** | | | | | | |
| Benin | 2 | 494 | 41.71 (18.58, 64.83) | 268.8763 | 96.6 | 0.000 |
| Burkina Faso | 3 | 627 | 24.08 (16.51, 31.65) | 35.5310 | 79.6 | 0.007 |
| Cameroon | 1 | 110 | 72.73 (64.41, 81.05) | 0.0000 | ----- | ------- |
| DR. Congo | 1 | 781 | 80.30 (77.51, 83.09) | 0.0000 | ----- | ------- |
| Ethiopia | 9 | 2817 | 69.21 (51.94, 86.49) | 693.3441 | 99.3 | 0.000 |
| Ivory Coast | 1 | 385 | 58.70 (53.78, 63.62) | 0.0000 | ----- | ------- |
| Nigeria | 2 | 966 | 46.02 (40.96, 51.09) | 8.0195 | 59.3 | 0.117 |
| Rwanda | 1 | 384 | 9.60 (6.65, 12.55) | 0.0000 | ----- | ----- |
| Tanzania | 2 | 604 | 37.57 (24.24, 50.89) | 84.5680 | 91.4 | 0.001 |
| Zambia | 1 | 411 | 5.87 (3.60, 8.14) | 0.0000 | -------- | ----- |
| **By Sub-African region** | | | | | | |
| East Africa | 13 | 4216 | 54.89 (34.64, 75.14) | ------- | 99.7 | 0.000 |
| West Africa | 8 | 2472 | 38.37 (28.67, 48.08) | 188.0303 | 96.2 | 0.000 |
| Central Africa | 2 | 891 | 77.57 (70.45, 84.70) | 18.6245 | 65.0 | 0.091 |
| **By study design** | | | | | | |
| Cross-section | 21 | 7012 | 52.10 (37.87, 66.34) | ----------- | 99.6 | 0.000 |
| Prospective cohort | 2 | 567 | 39.53 (1.90, 77.17) | 729.6868 | 99.0 | 0.000 |
| **By serological analysis done test** | | | | | | |
| Enzyme-linked fluorescent assay | 2 | 1166 | 69.60 (48.43, 90.77) | 229.1189 | 98.2 | 0.000 |
| Latex agglutination test | 5 | 1530 | 67.32 (39.19, 95.44) | ---------- | 99.5 | 0.000 |
| Enzyme-linked immunosorbent assay | 15 | 4472 | 46.11 (31.98, 60.25) | 772.4974 | 99.2 | 0.000 |
| Rapid test cassettes by CTK Biotech | 1 | 411 | 5.87 (3.60, 8.14) | 0.000 | ------- | -------- |
| **Total** | **23** | **7579** | 51.01 (37.66, 64.34) | ----------- | 99.6 | 0.000 |

** I-squared: the variation in ES attributable to heterogeneity)

**Table 3. Meta-regression analysis of the different study-level sources of heterogeneity for meta-analysis of the prevalence of toxoplasmosis among pregnant women in Africa.**

| Subgroup | Variables | Coefficient | P-value |
|---|---|---|---|
| **Countries of Africa** | Benin | 35.87 | 0.177 |
| | Burkina Faso | 18.02 | 0.460 |
| | Cameroon | 66.86 | 0.040 |
| | DR. Congo | 74.43 | 0.023 |
| | Ethiopia | 63.34 | 0.012 |
| | Ivory Coast | 52.83 | 0.092 |
| | Nigeria | 40.42 | 0.131 |
| | Rwanda | 3.71 | 0.899 |
| | Tanzania | 31.80 | 0.779 |
| **By Sub-African region** | East Africa | -23.68 | 0.289 |
| | West Africa | -38.23 | 0.080 |
| **By study design** | Cross section | 12.57 | 0.547 |
| **By serological analysis done test** | Enzyme-linked fluorescent assay | 63.67 | 0.052 |
| | Latex agglutination test | 40.24 | 0.136 |
| | Enzyme-linked immunosorbent assay | 61.44 | 0.037 |

 

a Joanna Briggs institute critical appraisal tool for prospective cohort and cross-sectional studies checklists [38].

The main aim of this meta-analysis is to estimate the prevalence of toxoplasmosis infection among pregnant women who attended an antenatal care unit in health facilities in Africa. This review included a total of 23 studies done in a facility-based setting in different regions of Africa. From the included studies, 21 were institutional-based cross-sectional studies [15–35], whereas 2 studies were institutional-based prospective cohort studies [36, 37]. The reports of all studies were based on a laboratory test to detect the presence of anti-toxoplasma antibodies in the blood samples during antenatal care follow-up based on international standards of tests [15–37]. For the laboratory tests, they applied four different types of serological tests in studies with blood samples; each study made use of blood samples for IgG and IgM anti-T. gondii specific antibodies whether positive or not. Based on these different serological test, 15 studies used enzyme-linked immunosorbent assay [15, 18, 22, 25–27, 29–35, 37]; 2 studies used enzyme-linked fluorescent assay [20, 36], 5 studies used latex agglutination test [16, 17, 21, 24, 28], and 1 study used rapid test cassettes by CTK Biotech [23] as per the standard operating procedures.

In this current analysis, the overall prevalence of Toxoplasmosis gondii varied across regions, with the lowest prevalence in the West African region and the highest prevalence in the Central African region. According to this meta-analysis, the overall pooled prevalence of toxoplasmosis infection among pregnant women was 51% in Africa. Hence, almost more than half of pregnant women identified with toxoplasmosis infection acquired it during pregnancy. Our finding shows a very high prevalence of toxoplasmosis infection in comparison with other systematic reviews and meta-analyses which previously published globally on acute toxoplasmosis among pregnant women [39]. The reason due to the globally conducted meta-analysis studies that used strict criteria for (seroconversion and low IgG gravidity) for the definition of acute toxoplasmosis of pregnant women had during gestation and not included the previous history of infection. Besides, these consequences are concurrent with the projected global incidence of congenital toxoplasmosis from a world health organization-supported study [40] that showed the highest incidence of congenital toxoplasmosis infection in some low-income African countries. Also, the current finding shows that toxoplasmosis infection needs serious attention among pregnant women in Africa because toxoplasmosis infection not treated at the early stages may affect many generations via transmission to the new-born, which may lead to foetal loss, neonatal death, and moderate to severe lifelong sequelae [5, 6, 40–42].

The strength of the present study: Authors used a protocol for search strategy, data abstraction, and conducted quality assessment by two independent investigators to reduce the possible appraiser bias; employed subgroup analysis based on the level of countries, study design, sub-African region, and serological test to identify the small study effect and the risk of heterogeneity in the study; and the quality of included studies was evaluated by five authors. However, our systematic review and meta-analysis have some limitations: a few studies were included in our subgroup analysis, which reduces the precision of our estimation; considerable high heterogeneity was identified among the studies; in this analysis, we did not consider studies written in other languages other than the English language, and also, our study did not include studies from the northern and southern regions of Africa; hence, it is difficult to generalise our finding to the whole of Africa.

## Conclusion

In summary, in our systematic review and meta-analysis, the pooled prevalence of toxoplasmosis among pregnant women in Africa was 51.01%. The pooled prevalence of toxoplasmosis

among pregnant women was 77.57% (highest) in Central Africa, 54.89% in East Africa, and 38.38% in West Africa. These figures show that there was a high number of toxoplasmosis infections among pregnant women in Africa. Therefore, we recommend that all countries emphasize the need to screen women during antenatal care, and if toxoplasmosis infection is detected, it should be treated early to reduce vertical transmission from mother to her unborn baby.

## Supporting information

**S1 File.**
(DOCX)

**S2 File.**
(DOCX)

**S3 File.**
(DOCX)

**S4 File.**
(XLSX)

## Acknowledgments

We would like to thank the College of Medicine and Health Sciences, Hawassa University (Ethiopia) for the non-financial support.

## Author Contributions

**Conceptualization:** Tamirat Tesfaye Dasa, Teshome Gensa Geta, Ayalnesh Zemene Yalew, Rahel Mezemir Abebe, Henna Umer Kele.

**Data curation:** Tamirat Tesfaye Dasa, Teshome Gensa Geta, Ayalnesh Zemene Yalew, Rahel Mezemir Abebe, Henna Umer Kele.

**Formal analysis:** Tamirat Tesfaye Dasa, Teshome Gensa Geta, Ayalnesh Zemene Yalew, Rahel Mezemir Abebe, Henna Umer Kele.

**Funding acquisition:** Tamirat Tesfaye Dasa, Teshome Gensa Geta, Ayalnesh Zemene Yalew, Rahel Mezemir Abebe, Henna Umer Kele.

**Investigation:** Tamirat Tesfaye Dasa, Teshome Gensa Geta, Ayalnesh Zemene Yalew, Rahel Mezemir Abebe, Henna Umer Kele.

**Methodology:** Tamirat Tesfaye Dasa, Teshome Gensa Geta, Ayalnesh Zemene Yalew, Rahel Mezemir Abebe, Henna Umer Kele.

**Project administration:** Tamirat Tesfaye Dasa, Teshome Gensa Geta, Ayalnesh Zemene Yalew, Rahel Mezemir Abebe, Henna Umer Kele.

**Resources:** Tamirat Tesfaye Dasa, Teshome Gensa Geta, Ayalnesh Zemene Yalew, Rahel Mezemir Abebe, Henna Umer Kele.

**Software:** Tamirat Tesfaye Dasa, Teshome Gensa Geta, Ayalnesh Zemene Yalew, Rahel Mezemir Abebe, Henna Umer Kele.

**Supervision:** Tamirat Tesfaye Dasa, Teshome Gensa Geta, Ayalnesh Zemene Yalew, Rahel Mezemir Abebe, Henna Umer Kele.

**Validation:** Tamirat Tesfaye Dasa, Teshome Gensa Geta, Ayalnesh Zemene Yalew, Rahel Mezemir Abebe, Henna Umer Kele.

**Visualization:** Tamirat Tesfaye Dasa, Teshome Gensa Geta, Ayalnesh Zemene Yalew, Rahel Mezemir Abebe, Henna Umer Kele.

**Writing – original draft:** Tamirat Tesfaye Dasa, Teshome Gensa Geta, Ayalnesh Zemene Yalew, Rahel Mezemir Abebe, Henna Umer Kele.

**Writing – review & editing:** Tamirat Tesfaye Dasa, Teshome Gensa Geta, Ayalnesh Zemene Yalew, Rahel Mezemir Abebe, Henna Umer Kele.

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
