## [Decision Letter · Decision Letter 0]

2 Mar 2021

PONE-D-20-27628

Toxoplasmosis infection among pregnant women in Africa: A systematic review and meta-analysis

PLOS ONE

Dear Dr. Dasa,

Thank you for submitting your manuscript to PLOS ONE. After careful consideration, we feel that it has merit but does not fully meet PLOS ONE’s publication criteria as it currently stands. Therefore, we invite you to submit a revised version of the manuscript that addresses the points raised during the review process.

Please attend to all the comments that have been provided by the reviewer.

This is an important study that can improves our understanding of toxoplasmosis in Africa. However, in its current form it is not easy to read because of the poor English. I suggest that the authors seek the services of a commercial editing service to help improve the writing.

We look forward to receiving your revised manuscript.

Kind regards,

Martin Chtolongo Simuunza, PhD

Academic Editor

PLOS ONE

Journal Requirements:

5. Thank you for submitting the above manuscript to PLOS ONE. During our internal evaluation of the manuscript, we found significant text overlap between your submission and the following previously published works, some of which you are an author.

- https://journals.plos.org/plosone/article?id=10.1371%2Fjournal.pone.0211764

- https://contraceptionmedicine.biomedcentral.com/articles/10.1186/s40834-019-0095-z

- https://www.sciencedirect.com/science/article/pii/B978012369542050026X?via%3Dihub

- https://www.euro.who.int/__data/assets/pdf_file/0011/294599/Factsheet-Toxoplasmosis-en.pdf

Please revise the manuscript to rephrase the duplicated text, cite your sources, and provide details as to how the current manuscript advances on previous work. Please note that further consideration is dependent on the submission of a manuscript that addresses these concerns about the overlap in text with published work.

Reviewers' comments:

Reviewer's Responses to Questions

**Comments to the Author**

1. Is the manuscript technically sound, and do the data support the conclusions?

Reviewer #1: Yes

2. Has the statistical analysis been performed appropriately and rigorously? 

Reviewer #1: Yes

3. Have the authors made all data underlying the findings in their manuscript fully available?

Reviewer #1: No

4. Is the manuscript presented in an intelligible fashion and written in standard English?

Reviewer #1: Yes

5. Review Comments to the Author

Reviewer #1: 1- In line 41 You have indicated that you have presented the Odds ratio with a 95% confidence interval of meta-analysis the random effect model for your pooled prevalence however the Odds ratio have be mention in your abstract only and nothing about it in any part of your text. You need to elaborate on the clearly in your manuscript text.

2- It good that the authors have included 23 studies in the analysis however the authors could discuss and reveal the biases, strengths, and weaknesses of those included studies.

3- In several places throughout the characteristics of included studies, prevalence of toxoplasmosis of pregnant women in Africa and meta-regression analysis for sources of heterogeneity sections, " 4 serological analysis tests" has been mentioned as a methods to test the presence of anti- Toxoplasmosis gondii antibodies. However, nothing has been mentioned about specificity or sensitivity of these tests or what is the source of differences between them. I suggest the authors could elaborate on that more. This increases the clarity of the sources of heterogeneity.

4- Meta-regression analysis for sources of heterogeneity and discussion give little information on the heterogeneity of pooled prevalence. It will be more useful if the authors can provide more details on source for heterogeneity and reveal the biases, strengths, and weaknesses of existing studies.

5- The authors have been able to make a balance between finding of the included studies that are similar and appropriate to combine without becoming too focused, also authors have been able to avoid a study population that is too narrow. However, The author need to elaborate more on validity and reliability of their study inference taking in consideration that the other studies written not by the English language were not considered in this analysis thin the studies from the northern and southern region of Africa not included. Pleas address the generalizable in more detail.

6. PLOS authors have the option to publish the peer review history of their article (what does this mean?). If published, this will include your full peer review and any attached files.

Reviewer #1: **Yes: **Abdullah Nagi Alosaimi

---

## [Author Response · Author response to Decision Letter 0]

24 May 2021

For reviewers, we have responded all comments accord to given comments of the reviewer. We want to appreciated our reviewers for their commitments.

---

## [Decision Letter · Decision Letter 1]

23 Jun 2021

Toxoplasmosis infection among pregnant women in Africa: A systematic review and meta-analysis

PONE-D-20-27628R1

Dear Dr. Dasa,

We’re pleased to inform you that your manuscript has been judged scientifically suitable for publication and will be formally accepted for publication once it meets all outstanding technical requirements.

Kind regards,

Martin Chtolongo Simuunza, PhD

Academic Editor

PLOS ONE

Additional Editor Comments (optional):

Reviewers' comments:

Reviewer's Responses to Questions

**Comments to the Author**

1. If the authors have adequately addressed your comments raised in a previous round of review and you feel that this manuscript is now acceptable for publication, you may indicate that here to bypass the “Comments to the Author” section, enter your conflict of interest statement in the “Confidential to Editor” section, and submit your "Accept" recommendation.

Reviewer #1: All comments have been addressed

2. Is the manuscript technically sound, and do the data support the conclusions?

Reviewer #1: Yes

3. Has the statistical analysis been performed appropriately and rigorously? 

Reviewer #1: Yes

4. Have the authors made all data underlying the findings in their manuscript fully available?

Reviewer #1: Yes

5. Is the manuscript presented in an intelligible fashion and written in standard English?

Reviewer #1: Yes

6. Review Comments to the Author

Reviewer #1: It was good that all data are fully available without restriction and the authors have declared that no competing interests exist.

7. PLOS authors have the option to publish the peer review history of their article (what does this mean?). If published, this will include your full peer review and any attached files.

Reviewer #1: **Yes: **Abdullah Nagi Alosaimi

---

## [Editor Report · Acceptance letter]

7 Jul 2021

PONE-D-20-27628R1 

Toxoplasmosis infection among pregnant women in Africa: A systematic review and meta-analysis 

Dear Dr. Dasa:

I'm pleased to inform you that your manuscript has been deemed suitable for publication in PLOS ONE. Congratulations! Your manuscript is now with our production department. 

Kind regards, 

on behalf of

Dr. Martin Chtolongo Simuunza 

Academic Editor

PLOS ONE